# Dose-Dependent Effects of Amino Acids on Clinical Outcomes in Adult Medical Inpatients Receiving Only Parenteral Nutrition: A Retrospective Cohort Study Using a Japanese Medical Claims Database

**DOI:** 10.3390/nu14173541

**Published:** 2022-08-27

**Authors:** Kosei Takagi, Kenta Murotani, Satoru Kamoshita, Akiyoshi Kuroda

**Affiliations:** 1Department of Gastroenterological Surgery, Okayama University Graduate School of Medicine, Dentistry, and Pharmaceutical Sciences, 2-5-1 Shikata-Cho, Kita-Ku, Okayama-City 700-8558, Okayama, Japan; 2Biostatistics Center, Kurume University, 67 Asahi-Machi, Kurume-City 830-0011, Fukuoka, Japan; 3Medical Affairs Department, Research and Development Center, Otsuka Pharmaceutical Factory, Inc., 2-9 Kanda Tsukasa-Machi, Chiyoda-Ku, Tokyo 101-8535, Japan; 4Research and Development Center, Otsuka Pharmaceutical Factory, Inc., 1-1 Kanda Ogawa-Machi, Chiyoda-Ku, Tokyo 101-0052, Japan

**Keywords:** parenteral nutrition, amino acids, medical inpatient, clinical outcomes, real-world data

## Abstract

The majority of inpatients requiring parenteral nutrition (PN) do not receive adequate amino acid, which may negatively impact clinical outcomes. We investigated the influence of amino acid doses on clinical outcomes in medical adult inpatients fasting >10 days and receiving only PN, using Japanese medical claims database. The primary endpoint was in-hospital mortality, and the secondary endpoints included deterioration of activities of daily living (ADL), intravenous catheter infection, hospital readmission, hospital length of stay (LOS), and total medical costs. Patients were divided into four groups according to their mean prescribed daily amino acid doses from Days 4 to 10 of fasting: Adequate (≥0.8 g/kg/day), Moderate (≥0.6–<0.8 g/kg/day), Low (≥0.4–<0.6 g/kg/day), and Very low (<0.4 g/kg/day). Multivariate logistic or multiple regression analyses were performed with adjustments for patient characteristics (total n = 86,702). The Adequate group was used as the reference in all analyses. For the Moderate, Low, and Very low groups, adjusted ORs (95% CI) of in-hospital mortality were 1.20 (1.14–1.26), 1.43 (1.36–1.51), and 1.72 (1.62–1.82), respectively, and for deterioration of ADL were 1.21 (1.11–1.32), 1.34 (1.22–1.47), and 1.22 (1.09–1.37), respectively. Adjusted regression coefficients (95% CI) of hospital LOS were 1.2 (0.4–2.1), 1.5 (0.6–2.4), and 2.9 (1.8–4.1), respectively. Lower prescribed doses of amino acids were associated with worse clinical outcomes including higher in-hospital mortality.

## 1. Introduction

Amino acids, the structural units of proteins, are known to play an important role in life-sustaining metabolic pathways, including those involved in maintaining muscle mass and immune function [1,2]. For that reason, adequate doses of amino acids should be provided to patients who are fasting and require parenteral nutrition (PN). Recent PN guidelines recommend providing an amino acid dose of 0.8 g/kg/day to 1.5 g/kg/day to these patients [3]. However, it has become apparent that, in clinical practice, patients often receive inadequate levels of nutrients in their PN. For example, a recent large-scale survey using a medical claims database revealed that the majority of hospitalized patients who had PN received inadequate doses of amino acids [4].

Beneficial effects of amino acids on survival have been reported for populations of elderly patients receiving PN [5,6]. However, few studies have been published looking at the impact of amino acids on populations of medical inpatients receiving PN [7]. Furthermore, investigations into the relationships between amino acids and a broader range of clinical outcomes, including mortality, complications, and medical costs, are lacking. Given that patients who require PN during hospitalization are typically already at increased risk of malnutrition and related sequelae leading to poor outcomes, and with the understanding that adequate amino acid doses in PN may have a positive impact on some outcomes, it would be valuable to obtain additional information about the relationships between amino acid doses and a range of clinical outcomes.

The aim of this study was to investigate the impact of prescribed doses of amino acids in PN on various clinical outcomes (i.e., in-hospital mortality, intravenous (IV) catheter infection, deterioration of activities of daily living (ADL), hospital read mission, and hospital length of stay (LOS)) and on medical costs in a large population of medical inpatients receiving only PN.

## 2. Materials and Methods

### 2.1. Study Design and Ethical Statements

We performed a retrospective cohort study using a medical claims database. This study was approved by the institutional ethics committee at the Okayama University Graduate School of Medicine, Dentistry, and Pharmaceutical Sciences (No. 2108-041) and the Kurume University Graduate School of Medicine (No. 21139), and it was registered at the University Hospital Medical Information Network Clinical Trial Registry (UMIN000044962).

This study was conducted in accordance with the Declaration of Helsinki and the Ethical Guidelines for Medical and Health Research Involving Human Subjects. This type of retrospective analysis using anonymized data does not require informed consent from the individual patients.

### 2.2. Data Source

The data was extracted from a medical claims database managed by the Medical Data Vision Co., Ltd. (MDV; Tokyo, Japan). As of October 2021, the MDV database contained data from 37.42 million patients at 451 hospitals, which accounted for approximately 26% of all Japanese medical institutions. The hospitals were all acute care hospitals that had advanced medical care capabilities and employed the diagnosis procedure combination/per-diem payment system. The database included information on the dates of patient admission and discharge, patient characteristics (i.e., age, sex, height, body weight, primary disease coded using the International Statistical Classification of Diseases and Related Health Problems, 10th Revision (ICD-10), comorbidities, ADL, and level of consciousness), diagnoses and treatments during hospitalization (i.e., diseases, medical treatments identified using Japan-specific medical claims codes, and prescribed drugs), number of beds in the hospital, types of admission, outcomes at discharge, and other data that were not used in our study.

### 2.3. Data Extraction

The extracted data included: age (categorized as 18–59, 60–69, 70–79, 80–89, or ≥90 years), sex, height, body weight, body mass index (BMI) (categorized according to the WHO classification [8] as <16.0, ≥16.0 to <18.5, ≥18.5 to <22.5, ≥22.5 to <25.0, ≥25.0 to <30.0, or ≥30.0), primary disease (sorted into 11 categories based on ICD-10 code), severity of comorbidities (determined using the Charlson Comorbidity Index (CCI) score in conjunction with the algorithm developed by Quan et al. [9] and categorized as 0, 1, 2, or ≥3), ADL (evaluated using the Barthel Index (BI) [10] and categorized as 0, 5–20, 25–40, 45–60, 65–95, or 100), and levels of consciousness (determined using the Japan Coma Scale (JCS) [11] and categorized as 0 (alert), 1–3 (not fully alert but awake without any stimuli), 10–30 (arousable with stimulation), or 100–300 (unarousable)), medical treatments (i.e., albumin infusion, blood transfusion, respirator use, dialysis, nutrition support, and/or rehabilitation) from Days 1 to 10, mean amino acid (g/kg/day) and energy (kcal/kg/day) doses on Days 4 to 10, number of beds in the hospital (categorized as <200, ≥200 to <500, or ≥500), year of admission (categorized as 2011–2012, 2013–2014, 2015–2016, 2017–2018, or 2019–2020), and type of admission (categorized as elective or emergency). For the study, Day 1 was regarded as the day that fasting started. Missing values for BI, JCS, and type of admission were categorized as “missing”. Total medical costs included medications, procedures, examinations, hospital charges, and other minor expenses.

### 2.4. Inclusion and Exclusion Criteria

This study included medical inpatients aged 18 years or older who were (1) admitted to the hospital between January 2011 and September 2020, (2) fasting (receiving no oral intake or enteral nutrition) for more than 10 consecutive days, and (3) managed with PN. Patients were excluded who (1) underwent surgery or were admitted to the intensive care unit from the day of hospital admission to the start of fasting, or (2) were managed with mean prescribed energy doses <10 kcal/kg/day or ≥30 kcal/kg/day on Days 4 through 10. The rationale for using Days 4 through 10 was that PN usually takes 3 to 4 days to reach the full target dose [4].

### 2.5. Exposure

The patients included in the study were divided into 4 groups according to their mean prescribed amino acid doses on Days 4 to 10: Adequate ≥0.8 g/kg/day, Moderate ≥0.6 to <0.8 g/kg/day, Low ≥0.4 to <0.6 g/kg/day, and Very low <0.4 g/kg/day.

### 2.6. Outcome Variables

The primary endpoint was in-hospital mortality. The secondary endpoints included deterioration of ADL, IV catheter infection during hospitalization, hospital readmission, hospital LOS, and total medical costs. Patients were considered to have deterioration of ADL when their BI score at discharge was lower than at admission. Hospital readmission was defined as being admitted within 30 days after discharge. Deterioration of ADL, hospital readmission, and hospital LOS were recorded only for patients who were discharged alive. Medical costs were calculated based on the Japanese yen and converted to the U.S. dollar (US$) using the 2020 annual exchange rate reported by the Organization for Economic Cooperation and Development (OECD) (US$1 = 107 Japanese yen) [12].

### 2.7. Statistical Analysis

Categorical variables were reported using frequencies and percentages, and continuous variables were described by medians and interquartile ranges (first quartile (Q1), third quartile (Q3)). Missing values were not imputed. Each variable was analyzed with either the Chi-square test or the Kruskal-Wallis test, with a two-sided significance level of 5%.

The association of in-hospital mortality with amino acid doses was analyzed with multivariate logistic regression using all extracted variables except amino acid doses as covariates. An adjusted odds ratio (OR) and 95% confidence interval (CI) were calculated for each amino acid dose group, with the Adequate (≥0.8 g/kg/day amino acid) group used as the reference. To eliminate the confounding of in-hospital mortality by energy doses, a subgroup analysis was also conducted. For this process, patients were further subdivided into 4 subgroups based on their mean prescribed energy doses on Days 4 to 10 (i.e., ≥10 to <15 kcal/kg/day, ≥15 to <20 kcal/kg/day, ≥20 to <25 kcal/kg/day, or ≥25 to <30 kcal/kg/day), and the association of in-hospital mortality with amino acid doses was reanalyzed with multivariate logistic regression for each energy dose subgroup.

Kaplan-Meier survival curves were generated for in-hospital mortality in each amino acid dose group, and the log-rank test was performed to investigate the differences between the groups. Furthermore, the adjusted hazard ratio (HR) and 95% CI for each amino acid dose group were calculated using the Cox proportional hazard model and the Adequate amino acid group as the reference. Patients discharged alive were censored on the day of discharge, and inpatients surviving for 180 days or longer were censored on Day 180.

Finally, either multivariate logistic regression analysis or multiple regression analysis was performed for the secondary outcome variables: deterioration of ADL, IV catheter infection, hospital readmission, hospital LOS, and total medical costs. The adjusted OR or regression coefficient with 95% CI was calculated for each amino acid dose group, with the Adequate (≥0.8 g/kg/day amino acid) group used as the reference. All statistical analyses were performed using SAS, version 9.4 (SAS Institute, Inc.; Cary, NC, USA).

## 3. Results

### 3.1. Patient characteristics

Of the 295,464 patients aged 18 years or older who were fasting for more than 10 consecutive days during hospitalization, 86,702 (33.4%) patients received PN and so met the inclusion criteria for the study (Figure 1). The characteristics of the patients in this study population are summarized in Table 1.

The proportion of the 86,702 patients in each amino acid dose group is summarized in Table 1. In the Adequate, Moderate, Low, and Very low amino acid groups, the median (Q1, Q3) of the mean prescribed amino acid doses on Days 4 to 10 was 0.92 (0.85, 1.00), 0.69 (0.64, 0.75), 0.51 (0.47, 0.56), and 0.29 (0.20, 0.36) g/kg/day, respectively. The median (Q1, Q3) of the mean prescribed energy doses on Days 4 to 10 was 20.0 (14.4, 25.2), 17.1 (12.1, 20.1), 14.1 (12.0, 16.1), and 12.1 (10.8, 14.7) kcal/kg/day, respectively.

### 3.2. Primary Endpoint: In-Hospital Mortality

The adjusted ORs for in-hospital mortality in the four amino acid dose groups are depicted in Figure 2. Adjusted ORs (95% CI) for in-hospital mortality were 1.20 (1.14–1.26) for the Moderate amino acid group, 1.43 (1.36–1.51) for the Low amino acid group, and 1.72 (1.62–1.82) for the Very low amino acid group, all relative to the Adequate amino acid group.

The Kaplan-Meier curves and adjusted HRs for in-hospital mortality, stratified by mean prescribed amino acid dose group, are depicted in Figure 3. The best survival was in the Adequate amino acid group and the worst survival was in the Very low amino acid group. Adjusted HRs (95% CI) for in-hospital mortality of the Moderate, Low, and Very low amino acid groups (relative to the Adequate amino acid group) were 1.10 (1.06–1.15), 1.26 (1.21–1.32), and 1.44 (1.37–1.51), respectively.

The adjusted ORs for in-hospital mortality in the 4 amino acid dose groups, within each of the 4 prescribed energy dose subgroups (i.e., ≥10 to <15 kcal/kg/day, ≥15 to <20 kcal/kg/day, ≥20 to <25 kcal/kg/day, or ≥25 to <30 kcal/kg/day of energy on Days 4 to 10), are depicted in Figure 4. Within each energy subgroup, with the Adequate amino acid group used as the reference, adjusted ORs for in-hospital mortality were always highest for the Very low amino acid groups, followed by the Low and then the Moderate groups.

### 3.3. Secondary Endpoint

There were significant differences between the four mean amino acid dose groups in deterioration of ADL, hospital LOS, and total medical costs (all *p* < 0.001), but not in IV catheter infections (*p* = 0.10) or hospital readmissions (*p* = 0.08) (Table 2).

The adjusted ORs or regression coefficients for deterioration of ADL, hospital LOS, and total medical costs were all significantly different among the four amino acid groups (Table 3). In contrast, the adjusted ORs for IV catheter infection and hospital readmission did not differ significantly between the four amino acid groups.

## 4. Discussion

This study involved 86,702 medical inpatients who were fasting and were prescribed PN, with the aim of investigating the effects of amino acid doses on clinical outcomes and total medical costs. We observed that lower prescribed amino acid doses in PN were significantly associated with higher odds of in-hospital mortality and deterioration of ADL, as well as longer hospital LOS and higher total medical costs. In contrast, we found no significant association between prescribed amino acid doses and the odds of IV catheter infection or hospital readmission.

Previous studies have also reported a positive effect of protein administration on clinical outcomes in critically ill patients [13,14,15]. In addition, positive correlations between amino acid intake and lower mortality have been demonstrated in older patients receiving PN and in patients with aspiration pneumonia [5,6]. However, the dose-dependent relationship between amino acids in PN and various clinical outcomes as well as medical costs has yet to be investigated.

This analysis is unique in that a large proportion of patients in the population were at high risk for poor outcomes. Patients at high risk included 65.3% who were 70 years or older, 35.4% who had a BMI lower than 18.5, 22.5% with a CCI of 3 or higher, 26.7% who required full assistance associated with reduced activity and prolonged bed rest (BI = 0), and 21.6% who were unconscious (JCS ≥ 1). Despite this, only 20,032 (23.1%) of the 86,702 patients that we studied were prescribed what are considered adequate daily doses of amino acids [3]. Furthermore, we found that many patients received low daily doses of energy. Possible explanations for this finding include that PN may have initially been given through peripheral rather than central venous lines, and that lipid injectable emulsions were not always included as part of PN [4].

Yet there is increasingly strong evidence that nutritional support has a positive effect on survival, particularly in malnourished medical inpatients [16,17]. Our findings provide additional evidence for this association. Not only did we observe that lower amino acid doses were associated with a higher risk of in-hospital mortality, but we also observed that the association between lower amino acid doses and in-hospital mortality remained even when patients were divided into different energy dose subgroups.

We also observed an association between lower amino acid doses and several other secondary clinical outcomes, including greater deterioration of ADL, longer hospital LOS, and higher total medical costs. The reduced physical activity and bedrest that occur during hospitalization may be accompanied by skeletal muscle loss, which may be further accelerated by various diseases, especially in the elderly [18,19,20]. Previous studies have also reported that skeletal muscle loss was associated with deterioration of ADL and increased mortality [21,22,23,24]. We suspect that the inadequate amino acid doses in PN prescribed for the hospitalized patients in our study may have hastened ongoing malnutrition and skeletal muscle loss, which then resulted in greater deterioration of ADL, longer hospital stays, and higher total medical costs for those patients. Our results also suggest that in future clinical outcomes studies involving medical inpatients, it may be beneficial to stratify findings according to the amino acid doses provided to them in their PN. Finally, our results also highlight the importance that clinicians should place on prescribing adequate amino acid doses for their medical inpatients receiving PN.

A strength of this study is that we examined the association of amino acid doses with in-hospital mortality in medical inpatients managed with PN using a medical claims database with a very large sample size. In addition, in order to mitigate the possibility of confounding by indication that can occur in retrospective studies, we calculated the ORs, HRs, and regression coefficients for our endpoints by adjusting for a broad range of patient demographic and clinical characteristics. Moreover, in order to mitigate potential confounding of the odds of in-hospital mortality by different energy doses included in PN, we performed an additional subgroup analysis of the relationship between amino acid doses and in-hospital mortality by also examining this relationship within four different prescribed energy subgroups. It was important to evaluate the effects of amino acid dose with the influence of energy dose having been eliminated because multi-chamber bags of PN, which combine amino acid and energy solutions into a single agent, are in common use [4]. Ultimately, this subgroup analysis confirmed a similar association between lower amino acid dose and higher risk of in-hospital mortality within each energy subgroup, further supporting the robustness of this relationship.

This study has several limitations. First, our findings are based on information registered in a medical claims database, which may contain missing data and entry errors. In addition, we coded primary diseases with ICD-10 retrospectively, which is inferior to characterizing diagnoses prospectively. However, we also used CCI as a retrospective measure of comorbidities, and this has been validated in Japan and is considered a reliable and accurate measure of comorbidities [25]. Second, despite the efforts to control bias, unknown or residual confounding factors may have existed. To address some of this potential confounding, we adjusted all regression analysis models for 17 demographic and clinical factors that could potentially have impacted the association between amino acid doses and clinical outcomes. However, other potential confounding factors such as disease severity and laboratory values were neither extracted from the database nor used in the study. Third, the doses of energy and amino acids used for our analysis were calculated from prescribed infusion preparations. Since data regarding the volume of infusion preparation used versus discarded was unavailable to us, the actual doses might have been even less than those used for our analysis. Fourth, we were limited in our ability to determine why patients received such low doses of amino acids and energy, because the database that we used did not include information about the indications for PN for individual patients. Nevertheless, our results still raise the possibility of significant gaps between recommended and prescribed amino acid and energy doses, which would be consistent with the “Knowledge-to-action” gap that has been reported by others in the field of clinical nutrition care [26,27]. Finally, although the study population included a broad range of medical inpatients admitted to acute care hospitals, our results may not be generalizable to chronic care hospitals.

## 5. Conclusions

Lower prescribed doses of amino acids in PN were associated with higher in-hospital mortality, greater deterioration of ADL, longer hospital LOS, and higher total medical costs for medical inpatients. In future clinical outcomes studies involving medical inpatients receiving PN, results should be analyzed according to the amino acid doses provided in the PN. Clinicians should also prioritize prescribing adequate amino acid doses for their medical inpatients receiving PN.

## Figures and Tables

**Figure 1 nutrients-14-03541-f001:**
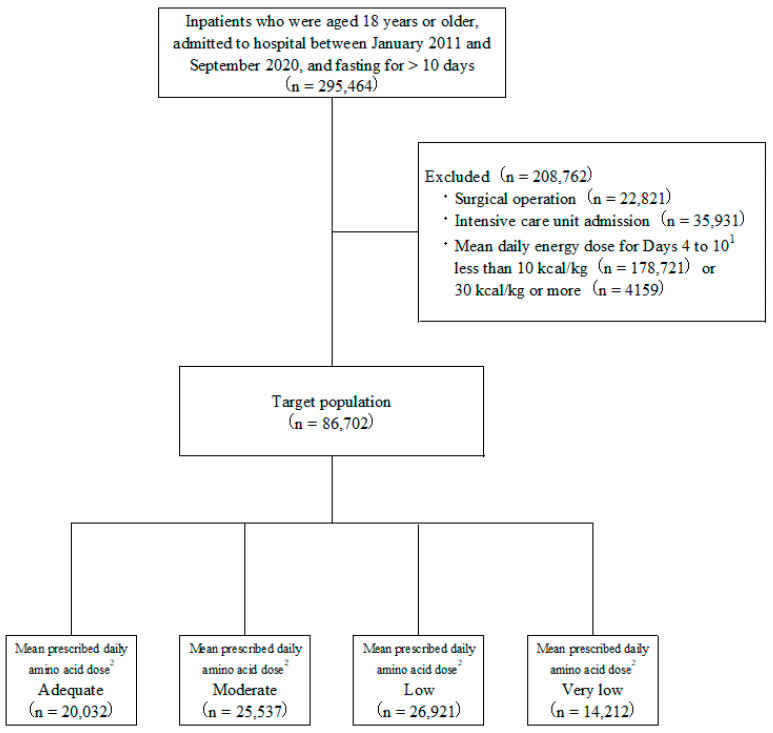
Study and patient disposition flowchart. ^1^ Day 1 regarded as the day that fasting started. ^2^ Classification according to mean prescribed amino acid dose on Days 4 to 10: Adequate ≥0.8 g/kg/day; Moderate ≥0.6, <0.8 g/kg/day; Low ≥0.4, <0.6 g/kg/day; and Very low <0.4 g/kg/day.

**Figure 2 nutrients-14-03541-f002:**
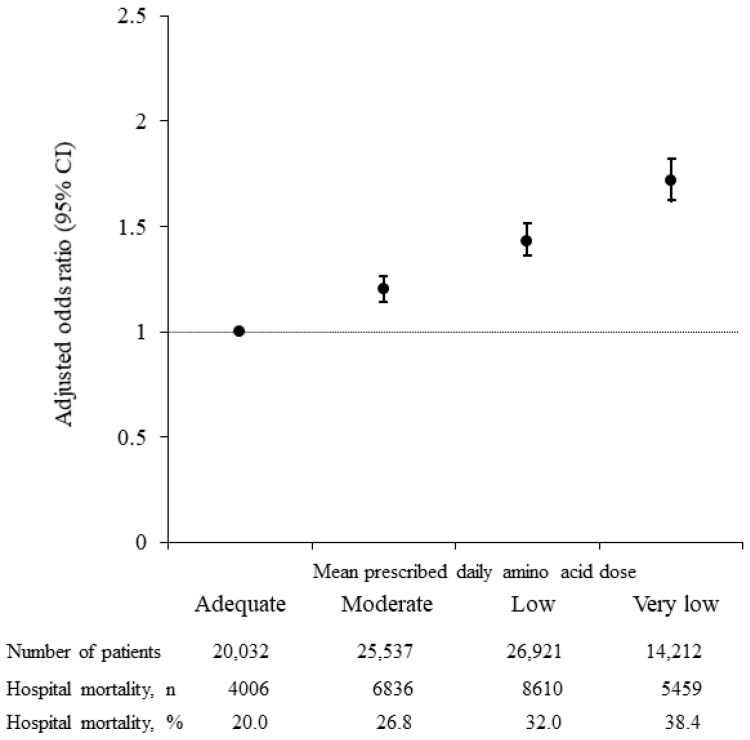
Adjusted odds ratios (ORs) and prevalence of in-hospital mortality 2011 through 2020, in patients grouped by mean prescribed amino acid dose on Days 4 to 10 (i.e., Adequate ≥0.8 g/kg/day; Moderate ≥0.6, <0.8 g/kg/day; Low ≥0.4, <0.6 g/kg/day; and Very low <0.4 g/kg/day); Day 1 regarded as day fasting started). Black dots indicate adjusted ORs for in-hospital mortality and vertical lines indicate 95% confidence intervals (CIs). With the Adequate amino acid dose group used as the reference, adjusted ORs (95% CIs) were: Moderate, 1.20 (1.14–1.26); Low, 1.43 (1.36–1.51); and Very low, 1.72 (1.62–1.82). Odds were adjusted for age, sex, body mass index, primary disease, Charlson Comorbidity Index, Barthel Index, Japan Coma Scale, medical treatment (albumin infusion, blood transfusion, respirator use, dialysis, nutrition support, rehabilitation) day of admission to Day 10, mean daily energy dose Days 4 to 10, number of beds in hospital, year of admission, and type of admission.

**Figure 3 nutrients-14-03541-f003:**
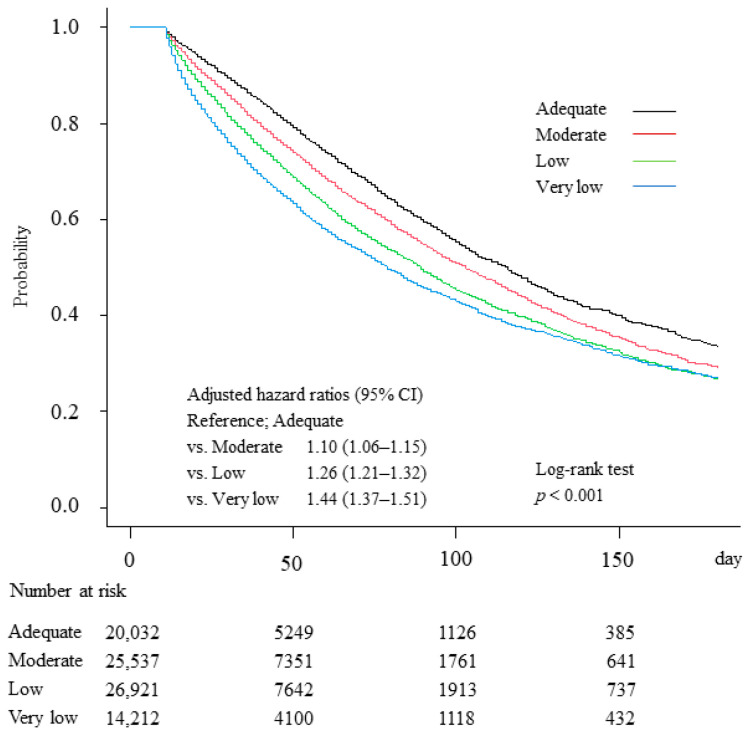
Kaplan-Meier curves and adjusted hazard ratios (HRs) with 95% confidence intervals (CIs) of in-hospital mortality in patients grouped by mean prescribed amino acid dose on Days 4 to 10 (i.e., Adequate ≥0.8 g/kg/day; Moderate ≥0.6, <0.8 g/kg/day; Low ≥0.4, <0.6 g/kg/day; and Very low <0.4 g/kg/day); Day 1 regarded as day fasting started. Adequate was used as the reference for HRs, which were adjusted for age, sex, body mass index, primary disease, Charlson Comorbidity Index, Barthel Index, Japan Coma Scale, medical treatment (albumin infusion, blood transfusion, respirator use, dialysis, nutrition support, rehabilitation) day of admission to Day 10, mean daily energy dose Days 4 to 10, number of beds in hospital, year of admission, and type of admission.

**Figure 4 nutrients-14-03541-f004:**
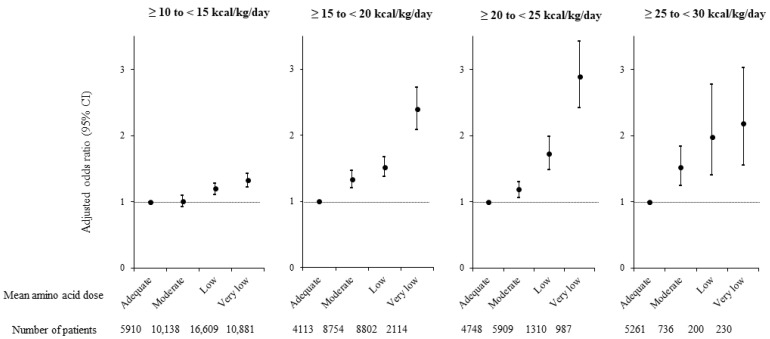
Adjusted odds ratios (ORs) of in-hospital mortality in patients subdivided by mean prescribed energy dose on Days 4 to 10 (i.e., ≥10 to <15 kcal/kg/day, ≥15 to <20 kcal/kg/day, ≥20 to <25 kcal/kg/day, or ≥25 to <30 kcal/kg/day of energy), and then further stratified by mean prescribed amino acid dose on Days 4 to 10 (i.e., Adequate ≥ 0.8 g/kg/day; Moderate ≥ 0.6, <0.8 g/kg/day; Low ≥ 0.4; <0.6 g/kg/day; and Very low < 0.4 g/kg/day); Day 1 regarded as day fasting started. Black dots indicate adjusted ORs for in-hospital mortality and vertical lines indicate 95% confidence intervals (CIs). Within each analysis, the Adequate amino acid dose group was used as the reference, and odds were adjusted for age, sex, body mass index, primary disease, Charlson Comorbidity Index, Barthel Index, Japan Coma Scale, medical treatment (albumin infusion, blood transfusion, respirator use, dialysis, nutrition support, rehabilitation) day of admission to Day 10, mean daily energy dose Days 4 to 10, number of beds in hospital, year of admission, and type of admission.

**Table 1 nutrients-14-03541-t001:** Distribution of demographic and clinical patient characteristics, 2011 through 2020, in patients stratified by mean prescribed daily amino acid dose in parenteral nutrition (PN).

*Characteristics*		Amino Acid Doses ^1^			
	Total (n = 86,702) n (%)	Adequate (n = 20,032) n (%)	Moderate (n = 25,537) n (%)	Low (n = 26,921) n (%)	Very Low (n = 14,212) n (%)
**Age**, years					
**18–59**	13,773 (15.9)	4296 (21.4)	3672 (14.4)	3992 (14.8)	1813 (12.8)
**60–69**	16,271 (18.8)	4325 (21.6)	4569 (17.9)	5051 (18.8)	2326 (16.4)
**70–79**	23,280 (26.9)	5452 (27.2)	6653 (26.1)	7418 (27.6)	3757 (26.4)
**80–89**	24,873 (28.7)	4582 (22.9)	7694 (30.1)	7960 (29.6)	4637 (32.6)
**>90**	8505 (9.8)	1377 (6.9)	2949 (11.5)	2500 (9.3)	1679 (11.8)
**Sex**					
**Male**	48,597 (56.1)	10,723 (53.5)	13,354 (52.3)	16,552 (61.5)	7968 (56.1)
**Female**	38,105 (43.9)	9309 (46.5)	12,183 (47.7)	10,369 (38.5)	6244 (43.9)
**Body mass index**, kg/m^2^					
**<16.0**	11,713 (13.5)	2064 (10.3)	3643 (14.3)	4119 (15.3)	1887 (13.3)
**≥16.0 to <18.5**	19,022 (21.9)	4278 (21.4)	5715 (22.4)	6128 (22.8)	2901 (20.4)
**≥18.5 to <22.5**	33,945 (39.2)	8220 (41.0)	10,043 (39.3)	10,200 (37.9)	5482 (38.6)
**≥22.5 to <25.0**	12,537 (14.5)	3057 (15.3)	3560 (13.9)	3751 (13.9)	2169 (15.3)
**≥25.0 to <30.0**	7957 (9.2)	1976 (9.9)	2207 (8.6)	2298 (8.5)	1476 (10.4)
**≥30**	1528 (1.8)	437 (2.2)	369 (1.4)	425 (1.6)	297 (2.1)
**Primary disease ^2^**					
**Digestive system malignancy**	22,853 (26.4)	6146 (30.7)	6994 (27.4)	7055 (26.2)	2658 (18.7)
**Hematological malignancy**	2865 (3.3)	594 (3.0)	843 (3.3)	953 (3.5)	475 (3.3)
**Other malignancy**	5577 (6.4)	1306 (6.5)	1692 (6.6)	1867 (6.9)	712 (5.0)
**Sepsis**	1494 (1.7)	240 (1.2)	407 (1.6)	483 (1.8)	364 (2.6)
**Coagulopathy disease**	827 (1.0)	121 (0.6)	190 (0.7)	287 (1.1)	229 (1.6)
**Cerebrovascular disease**	2952 (3.4)	565 (2.8)	933 (3.7)	899 (3.3)	555 (3.9)
**Cardiovascular disease**	3561 (4.1)	444 (2.2)	918 (3.6)	1160 (4.3)	1039 (7.3)
**Respiratory disease**	13,090 (15.1)	2020 (10.1)	4274 (16.7)	4391 (16.3)	2405 (16.9)
**Digestive system disease**	19,662 (22.7)	5813 (29.0)	5467 (21.4)	5303 (19.7)	3079 (21.7)
**Kidney/urinary tract disease**	2761 (3.2)	398 (2.0)	669 (2.6)	799 (3.0)	895 (6.3)
**Others**	11,060 (12.8)	2385 (11.9)	3150 (12.3)	3724 (13.8)	1801 (12.7)
**Charlson Comorbidity Index (CCI)**					
**0**	33,264 (38.4)	7920 (39.5)	9721 (38.1)	10,179 (37.8)	5444 (38.3)
**1**	3676 (4.2)	542 (2.7)	940 (3.7)	1124 (4.2)	1070 (7.5)
**2**	30,286 (34.9)	7431 (37.1)	9210 (36.1)	9222 (34.3)	4423 (31.1)
**≥3**	19,476 (22.5)	4139 (20.7)	5666 (22.2)	6396 (23.8)	3275 (23.0)
**Barthel Index (BI)**					
**100**	31,278 (36.1)	9477 (47.3)	8894 (34.8)	8995 (33.4)	3912 (27.5)
**65–95**	8177 (9.4)	1932 (9.6)	2393 (9.4)	2622 (9.7)	1230 (8.7)
**45–60**	5588 (6.4)	1175 (5.9)	1636 (6.4)	1815 (6.7)	962 (6.8)
**25–40**	3051 (3.5)	568 (2.8)	898 (3.5)	1004 (3.7)	581 (4.1)
**5–20**	5333 (6.2)	905 (4.5)	1611 (6.3)	1783 (6.6)	1034 (7.3)
**0**	23,117 (26.7)	3783 (18.9)	7140 (28.0)	7587 (28.2)	4607 (32.4)
**Missing**	10,158 (11.7)	2192 (10.9)	2965 (11.6)	3115 (11.6)	1886 (13.3)
**Japan Coma Scale (JCS)**					
**0**	65,633 (75.7)	16,529 (82.5)	19,081 (74.7)	20,107 (74.7)	9916 (69.8)
**1–3**	11,463 (13.2)	1938 (9.7)	3506 (13.7)	3803 (14.1)	2216 (15.6)
**10–30**	4655 (5.4)	762 (3.8)	1495 (5.9)	1438 (5.3)	960 (6.8)
**100–300**	2638 (3.0)	434 (2.2)	758 (3.0)	811 (3.0)	635 (4.5)
**Missing**	2313 (2.7)	369 (1.8)	697 (2.7)	762 (2.8)	485 (3.4)
**Medical treatment ^3^**					
**Albumin infusion**	9840 (11.3)	1661 (8.3)	2389 (9.4)	3294 (12.2)	2496 (17.6)
**Blood transfusion**	16,208 (18.7)	3125 (15.6)	4461 (17.5)	5321 (19.8)	3301 (23.2)
**Respirator use**	5146 (5.9)	528 (2.6)	1130 (4.4)	1883 (7.0)	1605 (11.3)
**Dialysis**	2464 (2.8)	100 (0.5)	211 (0.8)	602 (2.2)	1551 (10.9)
**Nutrition support**	6204 (7.2)	1281 (6.4)	1913 (7.5)	1981 (7.4)	1029 (7.2)
**Rehabilitation**	29,906 (34.5)	5450 (27.2)	9130 (35.8)	9938 (36.9)	5388 (37.9)
**Parenteral nutrition (PN) ^4^**					
**Amino acid**, g/kg/day, median (Q1,Q3)	0.61 (0.47, 0.79)	0.92 (0.85, 1.00)	0.69 (0.64, 0.75)	0.51 (0.47, 0.56)	0.29 (0.20, 0.36)
**Energy**, kcal/kg/day, median (Q1,Q3)	15.0 (12.1, 19.4)	20.0 (14.4, 25.2)	17.1 (12.1, 20.1)	14.1 (12.0, 16.1)	12.1 (10.8, 14.7)
**Number of hospital beds**					
**<200**	8844 (10.2)	1725 (8.6)	2634 (10.3)	2796 (10.4)	1689 (11.9)
**≥200 to <500**	47,890 (55.2)	10,690 (53.4)	14,019 (54.9)	14,997 (55.7)	8184 (57.6)
**≥500**	29,968 (34.6)	7617 (38.0)	8884 (34.8)	9128 (33.9)	4339 (30.5)
**Year of admission**					
**2011–2012**	7412 (8.5)	1997 (10.0)	2153 (8.4)	1998 (7.4)	1264 (8.9)
**2013–2014**	15,217 (17.6)	3850 (19.2)	4574 (17.9)	4238 (15.7)	2555 (18.0)
**2015–2016**	21,442 (24.7)	5167 (25.8)	6526 (25.6)	6339 (23.5)	3410 (24.0)
**2017–2018**	24,143 (27.8)	5343 (26.7)	7019 (27.5)	7943 (29.5)	3838 (27.0)
**2019–2020**	18,488 (21.3)	3675 (18.3)	5265 (20.6)	6403 (23.8)	3145 (22.1)
**Type of admission**					
**Elective**	47,395 (54.7)	11,996 (59.9)	13,897 (54.4)	14,606 (54.3)	6896 (48.5)
**Emergency**	39,233 (45.3)	8027 (40.1)	11,624 (45.5)	12,291 (45.7)	7291 (51.3)
**Missing**	74 (0.1)	9 (0.0)	16 (0.1)	24 (0.1)	25 (0.2)

^1^ Classification according to mean prescribed daily amino acid dose on Days 4 to 10: Adequate ≥0.8 g/kg/day; Moderate ≥0.6, <0.8 g/kg/day; Low ≥0.4, <0.6 g/kg/day; and Very low <0.4 g/kg/day; Day 1 regarded as day fasting started. ^2^ Based on International Statistical Classification of Diseases and Related Health Problems, 10th Revision (ICD-10). ^3^ Day of admission through Day 10. ^4^ Mean prescribed daily dose on Days 4 to 10; Day 1 regarded as day fasting started. Abbreviations: Q1, first quartile; Q3, third quartile.

**Table 2 nutrients-14-03541-t002:** Distribution of secondary endpoints and medical cost amounts, in patients grouped by mean prescribed daily amino acid dose in parenteral nutrition (PN).

*Secondary Endpoints*	Amino Acid Doses ^1^				*p*-Value
	Adequate (n = 20,032)	Moderate (n = 25,537)	Low (n = 26,921)	Very Low (n = 14,212)	
**Deterioration of ADL ^2^**, n (%)	1299 (9.5)	1960 (12.3)	2109 (13.5)	936 (12.8)	<0.001
**IV catheter infection**, n (%)	218 (1.1)	236 (0.9)	234 (0.9)	132 (0.9)	0.10
**Hospital readmission ^2^**, n (%)	1127 (7.0)	1383 (7.4)	1393 (7.6)	603 (6.9)	0.08
**Hospital LOS ^2^**, days, median (Q1, Q3)	33.0 (24.0, 49.0)	34.0 (24.0, 53.0)	34.0 (23.0, 53.0)	36.0 (24.0, 57.0)	<0.001
**Total medical costs**, US $, median (Q1; Q3)	$15,729 ($10,430; $23,486)	$16,037 ($10,617; $24,299)	$16,411 ($10,701; $25,561)	$16,804 ($10,766; $27,009)	<0.001

^1^ Classification according to mean prescribed daily amino acid dose on Days 4 to 10: Adequate ≥0.8 g/kg/day; Moderate ≥0.6, <0.8 g/kg/day; Low ≥0.4; <0.6 g/kg/day; and Very low <0.4 g/kg/day; Day 1 regarded as day fasting started. ^2^ Only includes patients discharged alive. Abbreviations: ADL, activities of daily living; IV, intravenous; LOS, length of stay; Q1, first quartile; Q3, third quartile.

**Table 3 nutrients-14-03541-t003:** Adjusted ^1^ odds ratios (ORs) or correlation coefficients and 95% confidence intervals (CIs) of secondary endpoints, in patients grouped by mean prescribed daily amino acid dose in parenteral nutrition (PN).

Secondary Endpoints	Amino Acid Doses ^2^			
	Adequate (n = 20,032)	Moderate (n = 25,537)	Low (n = 26,921)	Very Low (n = 14,212)
**Deterioration of ADL ^3^**	Reference	1.21 (1.11–1.32)	1.34 (1.22–1.47)	1.22 (1.09–1.37)
**IV Catheter infection**	Reference	1.06 (0.87–1.29)	1.15 (0.92–1.42)	1.21 (0.93–1.57)
**Hospital readmission ^3^**	Reference	1.10 (1.01–1.20)	1.11 (1.01–1.22)	1.07 (0.95–1.20)
**Hospital LOS ^3^**, days	Reference	1.2 (0.4–2.1)	1.5 (0.6–2.4)	2.9 (1.8–4.1)
**Total medical cost**, US $	Reference	$1664 ($1234–$2093)	$2505 ($2047–$2953)	$2486 ($1944–$3028)

^1^ Adjusted for age, sex, body mass index, primary disease, Charlson Comorbidity Index, Barthel Index, Japan Coma Scale, medical treatment (albumin infusion, blood transfusion, respirator use, dialysis, nutrition support, rehabilitation) day of admission to Day 10, mean daily energy dose Days 4 to 10, number of beds in hospital, year of admission, and type of admission. ^2^ Classification according to mean prescribed daily amino acid dose on Days 4 to 10: Adequate ≥0.8 g/kg/day; Moderate ≥0.6, <0.8 g/kg/day; Low ≥0.4; <0.6 g/kg/day; and Very low <0.4 g/kg/day; Day 1 regarded as day fasting started. ^3^ Only includes patients discharged alive. Abbreviations: ADL, activities of daily living; IV, intravenous; LOS, length of stay; Q1, first quartile; Q3, third quartile.

## Data Availability

Data described in the manuscript, code book, and analytic code will be made available upon request pending application and approval from Otsuka Pharmaceutical Factory, Inc.

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
