# Peer review of "Dose-Dependent Effects of Amino Acids on Clinical Outcomes in Adult Medical Inpatients Receiving Only Parenteral Nutrition: A Retrospective Cohort Study Using a Japanese Medical Claims Database"

_nutrients, 2022, doi:10.3390/nu14173541_

Round 1
Reviewer 1 Report
This is an interesting work correlating the grs of aminoacids prescribed to receive medical patients fasted for over 10d and receiving absolutely parenteral nutrition.
Ι would like the authors to clarify the reasons why some patients were prescribed smaller doses of amino acids. And, in case it is not possible to make an assumption about it in the discussion, and to comment about it on the limitations of the study
there are 1551 dialysis patients who received very low protein for for obvious reasons. in these patients the cause of increased mortality was the low protein or the worse medical status? What do you think for the opion to exclude those patients to whom low aminoacids had been prescribed for obvious reasons.
Author Response
August 24, 2022
Ms. Evelyn Ji
Assistant Editor
Nutrients
Dear Editor:
RE: manuscript number Nutrients-1878083
Dose-dependent effects of amino acids on clinical outcomes in adult medical inpatients receiving parenteral nutrition: A retrospective cohort study using a Japanese medical claims database.
Thank you for reviewing our manuscript. We are pleased that our manuscript was favorably reviewed and found to be potentially acceptable for publication pending revisions.
We thank the Reviewers for their valuable insights and comments, as these serve to further strengthen our manuscript.
As requested, we have provided below a point-by-point response to the Reviewer comments along with a description of the relevant changes made to the manuscript.
Sincerely,
On behalf of all coauthors,
Kosei Takagi, MD, PhD
Department of Gastroenterological Surgery, Okayama University Graduate School of Medicine, Dentistry, and Pharmaceutical Sciences, 2-5-1 Shikata-cho, Kita-ku, Okayama 700-8558, Japan
Tel: +81-86-223-7151; Fax: +81-86-221-8775; E-mail: kotakagi15@gmail.com
Reviewer #1:
This is an interesting work correlating the grs of amino acids prescribed to receive medical patients fasted for over 10d and receiving absolutely parenteral nutrition.
Comment 1:
I would like the authors to clarify the reasons why some patients were prescribed smaller doses of amino acids. And, in case it is not possible to make an assumption about it in the discussion, to comment about it on the limitations of the study.
Response:
Thank you for your important comment. Because the database we used did not include information regarding the reasons why some patients were prescribed smaller doses of amino acids, we were unable to provide an assumption in the discussion. As the reviewer 1 pointed out, we agree that this is a limitation of this study. Therefore, we have added this concern to the Discussion limitations section of the manuscript (page 12, line 359-369).
Comment 2:
there are 1551 dialysis patients who received very low protein for obvious reasons. in these patients the cause of increased mortality was the low protein or the worse medical status? What do you think for the option to exclude those patients to whom low amino acids had been prescribed for obvious reasons.
Response:
Thank you for your comment. Regarding distribution of patient characteristics depicted in Table 1, there are several differences, including primary diseases and medical treatment, between the groups stratified by amino acid doses as the reviewer 1 pointed out. Therefore, we performed multivariate logistic regression analysis, multiple regression analysis, or the Cox proportional hazard model analysis to calculate the ORs, regression coefficients, and HRs for our endpoints by adjusting for a broad range of patient demographic and clinical characteristics. The purpose of adjusting for these characteristics, including for dialysis, was to mitigate the risk that any one variable in a category of patients would bias or skew the overall clinical outcome results. And, we believe that the fact that our study included a wide range of medical inpatients is actually a noteworthy strength of this study. Regarding the option of excluding patients from the study who had low amino acid doses prescribed for specific reasons, we agree with the Reviewer that this could possibly yield some valuable information. However, as noted in a response above, the database that we used lacked specific data about individual patient indications for PN component dosing (which we have acknowledged above as being a limitation), and we would be concerned that excluding some groups of patients and not others, based on our limited understanding of all of each patient’s comorbidities and clinical condition, might potentially introduce unnecessary bias into our study.

Reviewer 2 Report
Thank you for the opportunity to review this interesting article. I really enjoyed reading it, however I do have several comments that should be addressed:
1. I think that the way you dealt with missing data should be described more in detail, in the table does the missing data mean NA? I think it should be stated that it is missing and not NA.
2. Also if you are doing a cost analysis it should be more thoroughly described what did you take into account?
3. In the results section reduce the text (especially in the 3.1 section) as it in most part just duplicate findings that are already written in the tables.
4. The discussion section could more discuss the findings with other similar studies, it would be very interesting to see a discussion about why so many patients received such low daily calorie doses (this was striking to me)
This article has great potential, however the aforementioned issues should be addressed.
Author Response
August 24, 2022
Ms. Evelyn Ji
Assistant Editor
Nutrients
Dear Editor:
RE: manuscript number Nutrients-1878083
Dose-dependent effects of amino acids on clinical outcomes in adult medical inpatients receiving parenteral nutrition: A retrospective cohort study using a Japanese medical claims database.
Thank you for reviewing our manuscript. We are pleased that our manuscript was favorably reviewed and found to be potentially acceptable for publication pending revisions.
We thank the Reviewers for their valuable insights and comments, as these serve to further strengthen our manuscript.
As requested, we have provided below a point-by-point response to the Reviewer comments along with a description of the relevant changes made to the manuscript.
Sincerely,
On behalf of all coauthors,
Kosei Takagi, MD, PhD
Department of Gastroenterological Surgery, Okayama University Graduate School of Medicine, Dentistry, and Pharmaceutical Sciences, 2-5-1 Shikata-cho, Kita-ku, Okayama 700-8558, Japan
Tel: +81-86-223-7151; Fax: +81-86-221-8775; E-mail: kotakagi15@gmail.com
Reviewer #2:
Thank you for the opportunity to review this interesting article. I really enjoyed reading it, however I do have several comments that should be addressed. This article has great potential, however the aforementioned issues should be addressed.
Comment 1:
I think that the way you dealt with missing data should be described more in detail, in the table does the missing data mean NA? I think it should be stated that it is missing and not NA.
Response:
Thank you for your comments. In the original manuscript, we showed missing values for BI, JCS, and type of admission as “unknown” categories. In the revised manuscript, we have defined missing values for BI, JCS, and type of admission as “missing” categories (page 3, line 110). In addition, we have listed missing data as “missing” in Table 1, as the reviewer has suggested.
Comment 2:
Also if you are doing a cost analysis it should be more thoroughly described what did you take into account?
Response:
Thank you for your comments. In the present study, total medical costs included medications, procedures, examinations, hospital charges, and other minor expenses. In the revised manuscript, we have more thoroughly described what was taken into account (page 3, line 110-111), as the reviewer has suggested.
Comment 3:
In the results section reduce the text (especially in the 3.1 section) as it in most part just duplicate findings that are already written in the tables.
Response:
Thank you for your comments. As the reviewer has suggested, we have reduced the amount of text in the Results section. We have eliminated essentially all of the text pertaining to Table 1 in the first paragraph of the Results section (page 4, line 172-173). Similarly, we have eliminated some of the text pertaining to Table 1 in the second paragraph of the Results section (page 7, line 200-206). In addition, we have compressed and reduced the text pertaining to Table 3 towards the end of the Results section (page 10, line 261-265).
Comment 4:
The discussion section could more discuss the findings with other similar studies, it would be very interesting to see a discussion about why so many patients received such low daily calorie doses (this was striking to me)
Response:
Thank you for your comments. Regarding the findings of similar studies, previous investigations have reported a positive effect of protein administration on clinical outcomes in critically ill patients [13-15]. In addition, a positive relationship of amino acid intake with mortality has been demonstrated in older patients receiving PN and patients with aspiration pneumonia [5, 6]. However, the association of amino acid doses with various clinical outcomes as well as medical costs has not yet been investigated so far. Therefore, to our knowledge, this is the first study investigating the effects of amino acid doses on clinical outcomes and total medical costs using a medical claims database with a very large sample size. In the revised manuscript, we have added the text above about this issue to the Discussion section as suggested by the Reviewer (page 11, line 293-298).
Regarding the results that low daily doses of energy were prescribed in this study, this might be because PN was initially started through peripheral vein route instead of central vein route, and lipid injectable emulsions were not being used commonly as part of PN. In the revised manuscript, we have added text about this issue in the Discussion section, as suggested by the Reviewer (page 11, line 305-309).
Furthermore, we have also postulated in our Limitations section that another possible reason for the low energy doses might be a “Knowledge-to-action” gap that has been reported in the field of clinical nutrition care [26, 27]. It is possible that the differences between recommended and prescribed PN component doses in our study might have been related to this gap. Thus, we have also added this text to the Discussion section of the revised manuscript, as suggested by the Reviewer (page 12, line 359-369).

Round 2
Reviewer 2 Report
Thank you for the changes you have made to the manuscript, I have no further comments.